# Public perspectives on COVID-19 triage protocols for access to critical care in extreme pandemic context

Marie-Eve Bouthillier[1‡]*, Yanick Farmer[2‡], Claudia Calderon Ramirez[3], James Downar[4], Andrea Frolic[5], Joseph Dahine[6], Lucie Opatrny[7], Diane Poirier[8], Gina Bravo[9], Audrey L'Espérance[10], Antoine Payot[11], Peter Tanuseputro[12], Louis-Martin Rousseau[13], Vincent Dumez[14], Annie Descôteaux[15], Clara Dallaire[15], Karell Laporte[16], Nathalie Orr Gaucher[17‡]

1 Department of Family and Emergency Medicine and Office of Clinical Ethics, Faculty of Medicine, Université de Montréal, Montréal, Québec, Canada, 2 Department of Social and Public Communication, Faculty of Communication, Université du Québec à Montréal, Montréal, Québec, Canada, 3 Biomedical Sciences Program, Clinical Ethics, Faculty of Medicine, Université de Montréal, Montréal, Québec, Canada, 4 Department of Critical Care, The Ottawa Hospital, Division of Palliative Care, Department of Medicine, University of Ottawa, Ottawa, Ontario, Canada, 5 Program for Ethics and Care Ecologies (PEaCE), Hamilton Health Sciences, Hamilton, Ontario, Canada, 6 Department of Medicine and Medical Specialties, Faculty of Medicine, Université de Montréal, Montréal, Québec, Canada, 7 Executive Office and Administration, Faculty of Medicine, McGill University Health Centre, Montréal, Québec, Canada, 8 Centre Intégré de Santé et de Services Sociaux du Centre-sud de Montréal, Montréal, Québec, Canada, 9 Department of Community Health Sciences, Faculty of Medicine and Health Sciences, Université de Sherbrooke, Sherbrooke, Québec, Canada, 10 École nationale d'administration publique (ENAP), Montréal, Québec, Canada, 11 Office of Clinical Ethics, Faculty of Medicine, Université de Montréal, Montréal, Québec, Canada, 12 Department of Family Medicine and Primary Care, Li Ka Shing Faculty of Medicine, The University of Hong Kong, Pokfulam, Hong Kong SAR, China, 13 Faculty of Engineering, Montreal Polytechnic, Montréal, Québec, Canada, 14 Centre d'Excellence sur le Partenariat avec les Patients et le Public (CEPPP) Centre de Recherche du Centre Hospitalier de l'Université de Montréal (CRCHUM), Montréal, Québec, Canada, 15 Bureau du Patient Partenaire, Faculty of Medicine, Université de Montréal, Montréal, Québec, Canada, 16 Medical Residency Program, Faculty of Medicine, Université de Montréal, Montréal, Québec, Canada, 17 Research Centre of the Sainte-Justine University Hospital, Montréal, Québec, Canada

‡ MEB, YF and NOG are joint senior authors on this work.

* marie-eve.bouthillier@umontreal.ca

**Data Availability Statement:** Data cannot be shared publicly because the ethical approval does not allow the sharing of data (even anonymized) on

## Abstract

COVID-19 triage protocols are resource allocation processes to deal with the potential lack of resources in Intensive Care Units (ICU). They have given rise to numerous ethical issues and controversies. Among them is the fear that people will be denied access to ICU on the basis of judgments about their quality of life, social value, frailty or age. This online Democratic Deliberation (DD) with members of the public aimed to discover the necessary considerations and conditions that make triage protocols more acceptable to guide future decisions in terms of the values and criteria that must underpin triage protocols. We simultaneously conducted the online DD in Quebec and Ontario on May 28th and June 4th, 2022, among adults who do not work in the healthcare sector, recruited randomly among the members of the public registered on Leger Opinion poll website to favor sociodemographic diversity. Data was analyzed using thematic analysis. Among the participants who took part in the study, 27 participants were from Ontario and 20 from Quebec. Three main themes emerged: 1) Acceptance of the protocol and values, 2) Considerations to be integrated in

a public repository and the participants did not consent to the sharing of their data publicly. Researchers who meet the criteria for access to confidential data may contact Marie-Alexia Masella (marie-alexia.masella@umontreal.ca) at the Comité d'éthique de la recherche en sciences et en santé (CERSES) with questions about data access.

**Funding:** This study was supported by the Canadian Institutes of Health Research (CIHR). Funding Reference Number/ EG4-179448 awarded to Marie-Eve Bouthillier.

**Competing interests:** The authors have declared that no competing interests exist.

triage protocols, 3) Conditions which may favor a greater public acceptance of these protocols. Participants supported the idea of prioritizing patients with the best prognosis of survival under extreme conditions. The maximization of benefits was the most predominant approach. Participants considered that triage protocols are necessary to reduce arbitrariness in decision making and to facilitate these tragic decisions by health professionals.

## Introduction

The aim of a triage protocol is to allocate limited resources in the most efficient way possible, giving them to patients who are likely to benefit most [1]. The key elements of triage protocols in the context of the COVID-19 pandemic have been specified in recommendation guidelines issued by scientific societies [1–3]. The Ontario and Quebec COVID-19 adult triage protocols were first to be developed in Canada during the first pandemic wave [4, 5]. A protocol for the pediatric and neonatal population was also created in the province of Quebec but is not discussed in this article [6].

### Criteria and process of ICU triage

These protocols are developed for crisis situations ($\geq$ 200% of surge capacity), i.e. when demand far exceeds capacity. Several other strategies (e.g. contingency planning) are put in place before resorting to them [1]. Published protocols for COVID-19 describe numerous clinical criteria used to prioritize patients such as their comorbidities, functional status, probability of survival, life expectancy, short-term prognosis, medical urgency, and frailty [7–9]. These protocols apply to all patients requiring ICU, not just those with COVID-19 [10]. Protocols also embody underlying values and principles, which guide resource allocation decision-making [11, 12]. Among these fundamental ethical values, two stand out: maximization of benefits (utilitarianism) and equal treatment of patients (egalitarianism) [13]. Added to this is equity, necessary to "balance" the negative effects of a vision too focused on benefit maximization, which would have no regard for vulnerability [14]. Some stress the importance of taking cultural values into account, such as those of indigenous populations, and other country-specific situations [15]. In the USA, the pandemic has disproportionately affected disadvantaged groups such as people living in poverty, Black and Latin Americans and finally, indigenous communities. As such, White & Lo (2021) suggest that triage protocols incorporate mitigation strategies to reduce the risk of exacerbating racial and sociodemographic disparities [14].

In addition to clinical criteria and underlying values, additional triage criteria (tiebreakers) have been proposed. Though controversial and criticized, they aim to break ties between patients whose other triage parameters are clinically equivalent. Common tie-breaker criteria include: life cycle, multiplier effect/social utility/instrumental value, caregivers at risk, randomization, among others [16, 17].

The triage process involves several essential steps: 1) clinical assessment of patients using the clinical triage criteria, 2) prioritization of patients by triage teams who are not at the patient's bedside, 3) coordination of ICU beds and transfer of patients where resources are available. As such, this process requires coordination and leadership at every level of the healthcare system.

Triggering a triage protocol in no way relieves of the obligation to care for all patients in need [1, 18, 19]. The main characteristics of the protocols of both provinces are presented in Table 1.

**Table 1. Main characteristics of Quebec and Ontario triage protocols for ICU access in a pandemic context.**

| | | |
|---|---|---|
| **Underlying principles and values** | • Benefit maximization, Proportionality, Equity, Transparency and trust, Efficiency and sustainability | |
| **Approach to Critical Care Triage** | • Surge capacity 150%: preparation for Prioritization.<br>• Surge capacity 200% and more: Prioritization according to the 3 levels in relation to system oversaturation [1].<br>• Criteria to change level of triage: When requests for access to an ICU bed exceed the availability of beds (transfers are not possible anymore). | |
| **Prioritization Criteria** | • **Short Term Mortality Risk (STMR):** Risk of death in the 12 months following critical illness.<br>•Focus on mortality<br>*risk* at 12 months, <u>not</u> the estimated survival duration for an individual.<br>• Not based on estimated survival duration in the absence of critical illness. | **Level 1 Triage:** Short term mortality risk <80% prioritized.<br>People with STMR risk >80% should not have critical care initiated.<br>**Level 2 Triage:** Short term mortality risk <50% prioritized.<br>People with STMR risk >50% should not have critical care initiated.<br>**Level 3 Triage:** Short term mortality risk <30% prioritized.<br>People with STMR risk >30% should not have critical care initiated. |
| **Triage process** | **1) INDIVIDUALIZED ASSESSMENT AT BEDSIDE**<br>• Informed by published data or expert opinion- clinical indicators suggested (standardized STMR assessment form).<br>**2) SECOND OPINION BY CRITICAL CARE PHYSICIAN**<br>• Ontario: The most optimistic assessment prevails. Triage decisions are hospital based.<br>• Quebec: STMR Assessment form sent to triage committee (two doctors and an ethicist or manager) who revise them all and make triage decision. An ICU triage executive committee supervises the process at provincial level, coordinates hospital transfers, examines capacity and change of triage level. | **3) ADMIT FOR TRIAL OF CRITICAL CARE**<br>• Communicate with administrator on call.<br>• Communicate decisions to patient/family.<br>**4) ADMIT/TRANSFER/REMAIN ON WARD**<br>Provide medical therapy as indicated.<br>• Add comfort orders, provide palliation.<br>• Reassess if triage downgraded.<br>• The Chief medical officer communicates an update on triage decisions, capacity, and transfers to the provincial ICU executive committee for coordination (Quebec). |
| **Tiebreakers** | **Quebec**<br>• Life cycle (intergenerational equity)<br>• Frontline healthcare workers in close contact with patient (reciprocity)<br>• Randomization (egalitarian justice). | **Ontario**<br>• Randomization (egalitarian justice) |

## Ethical issues of ICU triage and the public's concerns

Once the COVID-19 protocols became public, they gave rise to numerous ethical questions [20]. Does triage save more lives than first-come-first-served or randomization? What values and principles should guide resource allocation? Which criteria and clinical tools should be used for assessing prognosis in the context of patient prioritization? Which tiebreaker criteria should be used the event of a tie?

Other ethical concerns are associated to prioritization protocols such as: 1) the moral distress experienced by both clinicians and families as a result of their application, particularly when clinical care is refused or withdrawn; 2) clinicians' conscientious objections, and 3) the lack of public consultation in their development. Concerns raised by the public include the fear that people will be denied access to ICU based on judgments regarding their quality of life or social value. Others worry that the use of certain prognostic tools and triage criteria, such as frailty scales and age, may be "intrinsically discriminatory". In fact, legal action has been taken by disabled people alleging that triage algorithms could disadvantage them [21]. In light of these concerns, some studies have surveyed the public's opinions on COVID-19 triage protocols [22–26]. While informative, these surveys do not offer participants the opportunity to be informed on these complex matters, nor do they offer them a platform where they can discuss and exchange their respective viewpoints. In fact, few studies have used deliberative methods with the general public on this subject matter [27, 28]. Kuylen et al. (2021) used online deliberative workshops to explore the general public's views in the UK regarding the use of age and frailty as triage criteria in the context of the COVID-19 pandemic [27].

Given the lack of public inclusion through deliberative methods on this controversial topic, we conducted an online democratic deliberation (DD) with members of the public in Quebec and Ontario, Canada to discover the essential considerations and conditions that can make

triage protocols more acceptable in terms of values and criteria to guide future decisions. The informed opinion of the public provides a non-expert viewpoint, identifies blind spots, and informs decision-makers regarding the relevance of developing such protocols.

## Materials and methods

### Online DD and overall design of the study

Based on dialogue and expert education, DD has proven to be effective in helping people with differing opinions to "reason together" and reach a collective recommendation on morally complex issues such as ICU triage [29, 30]. The use of DD on other ethical issues has shown that citizens can formulate basic moral principles, question them, recognize competing moral considerations, and provide convincing arguments to support their position [30, 31]. DD is also a source of innovation, as it generates rich ideas from the collective dialogue fueled by the diverse experiential knowledge of participants [32].

The research paradigm of this study is constructivist. Through the DD, we offered a space for co-construction between the participants, where they could share their perspectives, challenge them and learn from each other. They were able to reflect together on triage protocols, in terms of values, as well as considerations and conditions that are necessary to make them more acceptable [33].

Part of the online DD presented here aimed to answering the following research questions:

*What is the best way to allocate scarce resources in the context of an extreme COVID-19 pandemic (first come first serve, triage protocol, randomisation)? What are the considerations and conditions that would favor the acceptability of triage protocols for access to ICU?*

The online DD was held simultaneously in Quebec and Ontario, in French and English respectively. Day 1, in May 2022, included a training session. Day 2, in June 2022 was the deliberative session. The preparation and logistics of this deliberation were jointly carried out by the research team and the Institute du Nouveau Monde (INM) which is an independent, non-partisan organization dedicated to increasing citizen participation in democratic life. The study design overview and procedures are described in Table 2.

### Ethical approval

This study obtained the ethical approval of two Canadian universities: On March 15, 2022, by the Comité d'éthique de la recherche en science et en santé (CERSES) de l'Université de Montréal (# 2022–1466), and on March 28, 2022, by the Bureau d'éthique et d'intégrité de la recherche de l'Université d'Ottawa (#H-03-22-8010). Written consent was obtained from all participants. In the development of this study, the standards for reporting qualitative research were taken from the checklist: Standards for Reporting Qualitative Research (SRQR) [34].

### Data analysis

We conducted a thematic analysis on the transcribed verbatim using NVivo14 released 2023, (Lumivero). First, three researchers independently coded a sample of the transcripts to identify patterns. They then met to agree on a set of codes. Two coders independently coded another sample, met again with the principal investigator, to discuss and resolve discrepancies in coding. A final set of codes was refined after several iterations. Observation notes were reviewed to corroborate the themes found in the coding process. An intercoder-reliability calculation was

**Table 2. Overview of the online DD Design and procedures.**

| | |
|---|---|
| **Recruitment process** | **Criteria for selecting participants:** Individuals over 18 years of age, from Quebec and Ontario, who were not currently working in healthcare or social services (to promote diverse backgrounds), capacity to easily participate in the online sessions and with an access to the Internet. **Recruitment procedure:** Carried out randomly by the firm Leger Opinion among the members of the public registered on their poll website. Our collaborator Institute du Nouveau Monde (INM) sampled among these participants to ensure a diversity of participants considering the following criteria: origin, age, gender, educational level, income, language, physical/mental fitness, and ethnicity. This selective sample included 60 participants from both provinces: 30 from Quebec and 30 from Ontario. |
| **Data Collection** | **One day training session, May 28[th], 2022** **Goal:** To give participants necessary information on prioritization protocols and their issues and offer them the opportunity to question the experts about them. **Presenters:** Eight experts (4 in Quebec, 4 in Ontario): professionals specialized in adult and pediatric critical care, ethics, anthropology, professionals working in partnership with patients, and university professors. **Themes of the training sessions**: 1) What are the models for prioritization in ICU, what is a triage protocol and what are the criteria? 2) Pediatric triage protocols; 3) Ethical issues, underlying values, tiebreakers; 4) Issues of discriminations (age, disability, ethnicity, etc.) **Agenda**: From 8:30 AM to 5 PM **Progress:** The schedule was the same in Quebec (French expert) and Ontario (English expert). The presenters worked together to ensure the same content. After each of the experts' simultaneous presentations, participants were free to ask any questions they might have, until all questions had been answered. Afterwards, participants divided into sub-groups to continue the conversation on what they had just heard, returning afterwards to the plenary session. At the end of the day, the final plenary session served to conclude the training session. **Half day deliberation session, June 4[th], 2022** **Goal:** To collect qualitative data on what are the public perspectives on allocation of scarce ICU resources in an extreme pandemic and the considerations and conditions to make triage protocols acceptable. **Questions under deliberation:** What is the best way to allocate scarce resources in the context of an extreme COVID-19 pandemic (first come first serve, triage protocol, randomization)? What are the considerations and conditions that would favor the acceptability of triage protocols for access to ICU? Under what values? **Agenda**: From 8:30 AM to 12 PM **Progress:** First, participants met in plenary to review the training session and summarize what they had heard. They then divided into four sub-groups to deliberate for an hour and a half. After a break, they returned to the plenary and pooled their discussions to synthesize their perspectives on the questions put to them. |
| **Facilitators** | For the two days, two main facilitators, one for the French sessions and the other for the English sessions, oversaw the animation of plenary sessions. They were assisted by 14 other facilitators who took care of the small group sessions. |
| **Note takers** | A total of six note takers consisted of students and professionals in clinical ethics. A structured online observation sheet was provided to take notes during the deliberation session. One of them was also available to assist participants in case they did not feel comfortable during the process. |
| **Recording and Confidentiality** | The sessions were conducted through the Zoom platform. Participants' opinions and statements were recorded and transcribed Verbatim in English and French. Prior to the thematic analysis, the transcripts were de-identified to preserve the confidentiality of the participants. All data were securely stored in locked files and password-protected in the University of Montreal's OneDrive system. |

performed in NVivo to determine the percentage of agreement between coders which was > 97%, and the kappa index global was 0.8 (S1 Table).

## Results

A total of 47 participants were included in the online DD, 27 from Ontario and 20 from Quebec, with a diverse demographic representation. The groups were balanced in terms of age range and gender. However, there were no gender diverse participants. We observed that most of the participants came from large urban cities such as the Capitale Nationale and the Greater Toronto Area, but we had an overall representation of 12 regions in Quebec and 6 in Ontario. We had a very rich and diverse representation, which included the presence of multiple visible and non-visible minorities, as well as the participation of one First Nations members in each of the two provinces. Participants were able to self-identify as belonging to any one of the visible minority groups, a category composed of multiple visible minority groups, or not belonging to a visible minority. These groups are mutually exclusive. Some participants had minor functional limitations (Ontario only). In terms of education, the majority had a high school diploma or higher. As for occupation, half of participants were employed, few were self-employed, others were retired, and the rest were unemployed but doing other activities or

studying. Most participants had an income between C$30,000 and C$69,999, only a few had an income of $100,000 or more. Demographic variables of the deliberants are presented in detail in Table 3.

## Qualitative results

The results of the qualitative analyses reveal three main themes, which we will elaborate on using the sub-themes that emerged from the analysis: 1) Acceptance of the protocol and values, 2) Other criteria to be integrated in triage protocols, 3) Conditions which may favor a greater public acceptance of these protocols.

## Acceptance of the protocol and underlying values

**Avoiding the question.** At the beginning of the online DD, participants tried to sidestep the initial question asked, which was: *What approach would be best to allocate scarce resources in the context of an extreme COVID-19 pandemic (first come first serve, triage protocol or randomization)?* Their initial responses would consist of comments that highlighted the need to invest more in the health system to provide resources all patients who require care the context of a pandemic crisis. One participant said that because no ideal solution would be found, artificial intelligence should be used to make these decisions to reduce biases.

"...*I was not thinking of machine learning but perhaps artificial intelligence...I wonder if that is not the solution, we put that in the machine and then we let it go because immediately I think that someone who starts to have a sort of choice to make or a decision to make, I think that leaves room for biases.*" (QP5)

The research team had to insist that discussions should focus not on hypothetical measures, but on the reality of the situation where there are no more ICU resources and hard choices must be made.

**Great acceptance of the protocol.** Most of the participants supported the idea of having protocols in an extreme pandemic scenario to save more lives, decrease chaos during these difficult moments, and to facilitate decision making. They also found them necessary to focus on the common good and avoid bias judgment by health care providers:

"*I think they are necessary to support the caregivers, the doctors, the nurses because it's true that involuntarily... we'll always try to save the one who needs it the most, but here we really have to think about the whole community, and I think that the protocols help us to leave aside the emotion to save as much as possible.*" (QC12)

"...*it allows us to avoid the nurse putting the other person first and the other person wanting to come first, not because it's mean but because it's our survival instinct and collectively the survival moment must be supervised...*" (QC4)

"...*a protocol allows us to avoid the law of the jungle*" (QC18)

**Underlying values of ICU triage.** Participants were asked to express their views on the essential values that should underpin triage protocols. According to participants, maximizing benefits, translated as saving as many lives as possible, was essential and received majority support. However, other values received significant support from participants, notably fairness. It's interesting to note that equality, which could be materialized in the form of a lottery, was

**Table 3. Sociodemographic characteristics of the participants.**

| Variables | n (%) | Quebec | n (%) | Ontario |
|---|---|---|---|---|
| | **20 (43)** | | **27 (57)** | |
| **Gender** | | | | |
| Male | 9 (45) | | 14 (52) | |
| Female | 11 (55) | | 13 (48) | |
| Other | 0 | | 0 | |
| **Age** | | | | |
| 18–24 | 3 (15) | | 2 (7) | |
| 25–34 | 1 (5) | | 3 (11) | |
| 35–44 | 3 (15) | | 5 (19) | |
| 45–54 | 6 (30) | | 5 (19) | |
| 55–64 | 2 (10) | | 5 (19) | |
| 65–74 | 4 (20) | | 4 (15) | |
| $\geq$75 | 1 (5) | | 3 (11) | |
| **Region** | | | | |
| | 1 (5) | Bas-Saint-Laurent | 5 (19) | Centre |
| | 1 (5) | Saguenay–Lac-Saint-Jean | 4 (15) | East |
| | 4 (20) | Capitale-Nationale | 12 (44) | Great Toronto |
| | 1 (5) | Estrie | 1 (4) | Northeast |
| | 3 (15) | Montréal | 1 (4) | Northwest |
| | 1 (5) | Outaouais | 4 (15) | Southwest |
| | 1 (5) | Abitibi-Témiscamingue | | |
| | 1 (5) | Côte-Nord | | |
| | 1 (5) | Gaspésie–Îles-de-la-Madeleine | | |
| | 2 (10) | Lanaudière | | |
| | 3 (15) | Montérégie | | |
| | 1 (5) | Centre-du-Québec | | |
| **First Nations** | | | | |
| Yes | 1 (5) | | 1 (4) | |
| No | 0 | | 0 | |
| **Functional limitations** | | | | |
| Yes | 0 | | 6 (22) | |
| No | 20 (100) | | 21 (78) | |
| **Self-reported visible minority groups †** | | | | |
| Asian (e.g. Chinese, Korean, Japanese) | 0 | | 3 (11) | |
| South Asian (e.g., Indian, Pakistani) | 0 | | 2 (7) | |
| Latin American | 2 (10) | | 0 | |
| Afro American | 0 | | 2 (7) | |
| Arab | 0 | | 1 (4) | |
| Visible minority | 1 (5) | | 1 (4) | |
| Multiple visible minorities | 0 | | 1 (4) | |
| Not a visible minority | 16 (80) | | 17 (63) | |
| I prefer not to answer | 1 (5) | | 0 | |
| **Schooling** | | | | |
| Primary (7 years or less) | 0 | | 0 | |
| Secondary (general or vocational training) | 4 (20) | | 7 (26) | |
| College (pre-university training, technical) | 8 (40) | | 6 (22) | |
| University 1 (bachelor's degree) | 4 (20) | | 9 (33) | |

(*Continued*)

**Table 3.** (Continued)

| Variables | n (%) | Quebec | n (%) | Ontario |
|---|---|---|---|---|
| | **20 (43)** | | **27 (57)** | |
| University 2 (master's degree) | 3 (15) | | 4 (15) | |
| University 3 (doctorate) | 0 | | 1 (4) | |
| I prefer not to answer | 1 (5) | | 0 | |
| **Occupation** | | | | |
| I am studying or in training | 0 | | 1 (4) | |
| I am working | 13 (65) | | 11 (41) | |
| I am taking care of my children or a relative | 1 (5) | | 0 | |
| I am retired | 3 (15) | | 9 (33) | |
| I am unemployed | 2 (10) | | 3 (11) | |
| I am self-employed | 1 (5) | | 3 (11) | |
| **Income** | | | | |
| Less than 30,000$ | 4 (20) | | 4 (15) | |
| 30 000$ à 49 999$ | 6 (30) | | 5 (19) | |
| 50 000$ à 69 999$ | 3 (15) | | 8 (30) | |
| 70 000$ à 99 999$ | 5 (25) | | 5 (19) | |
| 100 000$ and more | 1 (5) | | 5 (19) | |

† In Canada, the Employment Equity Act defines visible minorities as "persons, other than Aboriginal peoples, who are non-Caucasian in race or non-white in colour". The visible minority population consists mainly of the following groups: South Asian, Chinese, Black, Filipino, Arab, Latin American, Southeast Asian, West Asian, Korean and Japanese. The 'Multiple visible minorities' category includes persons who provided two or more groups designated as visible minorities.

initially less supported, but acceptable as a last resort, since it's neutral, unbiased and easy to apply. However, participants did not consider this to be the preferred method for patient triage. In their view, fairness was important to "correct" the effects of prioritization based solely on the idea of saving as many lives as possible, which would disadvantage certain groups. Other values were named as important to uphold when developing a triage protocol. These included transparency, responsibility/accountability, dignity and respect. Tables 4–6 show the range of values that were discussed at the online DD.

## Other criteria to be considered in triage protocols

When asked about other criteria to consider when triaging patients, in addition to clinical criteria (giving priority to those with the best chance of survival), participants discussed several aspects that should be considered. However, none of them were unanimously accepted by the group. These included: personal responsibility for health, consideration for pregnant women and disadvantaged groups.

## Personal responsibility for health

Participants believed that people who are considered responsible for their own ill-health should not be given the same priority as those who are not. Consequently, vaccination status was one example that was much discussed during the deliberations.

> . . .So, people who were able to take the vaccine but decided not to, they are being reckless with everyone else. So are we going to save them when they chose not to do their best for the wellbeing of the population. But even with that question, First Nations, Inuit and Metis should be

**Table 4. Fundamental principle for prioritization in pandemic voiced by the public.**

| Principle | Arguments in Agreement | Arguments in Disagreement | Neutral or don't know |
|---|---|---|---|
| **Prioritization based on Maximization of benefits** | • *"...I think we're in an extreme situation where there would be a lot of deaths, I think I prioritize more who has the best chance of survival; it's to continue the evolution of humanity;...I think we should really follow the protocol according to the criteria of who would have the best chance of survival."* (QP8)<br>• *...we are looking at it in terms of maximizing the number of people who leave the hospital alive, then you can basically simplify the process by going back and saying, "Who has the best chance of survival?". And using that as the essential criteria that you look at when trying to do this.* (OP20)<br>• *This is my first time learning it and I am grateful for the opportunity to participate. What I think is the best is the survival rate of each individual when managing a hospital. I feel like this protocol can kind of get things running smoothly in the hospital without things getting too chaotic. The higher the survival rate of a person, that person should get the treatment.* (OP23)<br>• *I think we're in general agreement here that survivability is the first and most important factor.* (OP10)<br>• *I would go with the utilitarianism and benefit maximization: save as many lives as possible.* (OP6) | • *Some people can't accept those types of decisions...It is very hard to follow that ethically. So, we have to choose accordingly and based on the situation. Like with COVID, we don't know what started it. And it can happen again in the future or every year or some other kind of pandemic. So, we should be prepared, and the governments should be more focused on how we can prepare for this in the future. So, we should have more facilities and more staff so that we can tackle the issues and manage them well.* (OP14) | • *"...you know it's definitely hard to know because it hasn't been put in place, it's been studied but it hasn't been put in place yet."* (QP16)<br>• *"I honestly don't know, I don't know what more I can say, it's not...it's debatable because it depends on the situation, it depends on who the person is, it depends on their health status, it depends on a lot of different things so you can't make a black and white decision."* (QP6) |

*exempt from that question...So, I don't blame them for not trusting the government and not taking the vaccine. So, they should be exempt from that question.* (ON8)

*Well, it is pretty hard to have sympathy for people who come in with "self-inflicted" issues, and then expect them to be treated the same as everybody else. I really have a challenge with that.* (ON18)

Those who disagreed stated that we should not consider personal responsibility in allocation decisions, nor was it the role of doctors to carry out this type of investigation on people's behavior.

*"...why should the doctor know that, I mean the guy who was speeding there I mean there's his accident, he comes in on a gurney, he's probably unconscious, are we going to start investigating? I say the doctor, but there is a whole health care team; is it their job to know why there was an accident, why he is in this state? I don't think so, and it's better that it's like that."* (QC17)

*Likewise, we do not deny healthcare to drunk drivers. We just don't. And to suggest that failing to get the vaccine (and I'm fully vaccinated, 4 doses), is reckless and that people should be denied care, it is just wrong.* (ON10)

## Consideration for pregnant women and disadvantaged people

Pregnant women were of great concern to participants, considering that they are patients who need to be prioritized taking into account their social and instrumental value.

**Table 5. Prioritization based on justice voiced by participants.**

| Principles and values | Arguments in Agreement | Arguments in Disagreement | Neutral or don't know |
|---|---|---|---|
| **a) Equity or fairness** | • *I still stand with benefits maximization and equity.* (OP6) <br> • *...One thing that jumps out to me when you are part of a community and go to a hospital needing care is fairness. Fairness is very important.* (OP21) <br> • *"I agree with this information because I think that it is necessary to avoid discrimination, then it is an equity through all this."* (QP18) | | |
| **b) Egality Randomization** | • *"What appeals to me in the drawing of lots is precisely the principle of equality of opportunity which confirms the principle of equality..."* (QP17) <br> • *...I heard somebody say that the lottery system is unfair, and maybe I am just playing semantics there, but I think it is very fair. It is completely unbiased. We may not like it, and we may not think the results are justified, but it is a fair way of doing it because everybody has an equal chance.* (OP20) <br> • *I agree that the random selection should only come as the last thing to do. If you can't do it by considering all these other things, then yes it should come down to random selection. But that should be the last choice in my mind.* (OP5) <br> • *The lottery has to be at the bottom, yes.* (OP26) | • *"...For the lottery I consider that it is a little too hasty a decision to use this lottery to choose who should be treated or not; it seems to me that we should go further..."* (QP16) <br> • *I guess I'll go backwards. Unacceptable for me would be the lottery. I already explained why that to me sounds ridiculous. I don't think anyone would be happy with their life being dependent on a coin flip. There are so many other relevant criteria that could be used.* (OP8) <br> • *For me, survival is the main thing. And lottery is totally not acceptable.* (OP24) <br> • *It is a totally fair way of doing it, but then the results may not make sense. You may get someone who has a great chance of survivability versus someone who has a poor chance of survivability, and you are devoting all your resources to someone who is not going to have a positive outcome. And the coin flip chose them, which is totally fair, but it is arguably a bad decision.* (OP20) | • *It may be that at a certain point in the triage, where you maybe have 5 people with all the other factors coming out equal, but you don't have enough resources for the 5 people. Maybe we can do something random at that point as a tiebreaker. But I don't know about using anything random before that.* (OP9) |
| **"First come, first served" Principle** | • *To me, the best thing would be first come/first served and the survival rate.* (OP7) <br> • *...The deciding factor first comes/first served after the science has been evaluated.* (OP26) | • *The acceptable one is survivability. And first come/first serve is kind of in the middle, but more leaning towards unacceptable.* (OP23) <br> • *First come/first serve is what I would except at McDonalds, not at my local hospital.* (OP10) <br> • *First come/first serve is pretty much unacceptable because we have more important criteria. And the preferred one would be survivability.* (OP8) <br> • *I guess for me, in the case of a critical care triage situation, I don't like first come/first serve. That actually feels more unfair to me. Looking at the bigger picture of whose lives can be saved and things, is more equitable than saying "oh, well they're here first". And again, that also exacerbates the access inequity.* (OP9) | • *I think that first come/first served is a good way to treat people when they all have the same sickness. If they all have the same severity of COVID, then we would treat them as first come/first served. But if you have someone who just has a cough and then you have someone who needs life support, then the person who needs life support would have to go first.* (OP27) <br> • *Yes, first is the first come/first served. And then the chance of survival. But in many cases, we're talking about a large number of patients coming in at the same time. Even if we do initial scoring, it is still going to be very hard to choose.* (OP7) |

*For social value, you really need to consider the fact that there are 2 lives rather than 1. And one of those lives hasn't even begun yet. So, they have everything to experience. And you also have a caregiver value in there.* (ON26)

*However, I do want to have pregnant women jump the cue. If the survivability is the same, I do think that pregnant women (two lives) should take priority.* (ON7)

Participants from Ontario were primarily concerned about the care provided to Indigenous Peoples, indicating that systemic discrimination of this group has resulted in numerous health and social disparities. For these reasons they should be prioritized. However, one participant

**Table 6. Other principles and values considered by the public.**

| Transparency | Responsibility /accountability | Solidarity | Dignity/respect |
|---|---|---|---|
| • I think fairness, transparency, and trust. (OP12)<br>• "The word transparency, in fact, is what appeals to me a lot. Transparency and openness. Transparency because, precisely, before such a protocol is put in place, it has to be really done, why it is done, why we are heading towards it, what is the motivation behind all that. . . the reasons and all that, I think that it has to be known,. . . and it takes a certain openness to accept such a protocol as well, to accept that such a person is chosen instead of another, but of course it has to be based on intelligent criteria." (QP8)<br>• Transparency is very important to me too. You want to know what the decision is, why it was, and make sure that the system was working. . .That sort of thing is crucial, particularly in these pandemic times. (OP21) | • Definitely responsibility and accountability, especially from the government. They are pretty much the reason we are having this discussion. . . (OP8)<br>• The improvement of the health of the population and the common good. These should be considered for this topic that we are discussing. And other one would be responsibility and accountability. That comes from all the sectors–from the government, governing bodies, the hospitals, everybody. (OP24)<br>• Hard to pick but I would say equality, responsibility and accountability, and improvement of the health population. (OP3) | • I totally do. It should not be just researchers and academics and small groups. It is an issue that everybody should be thinking about. . . (OP5)<br>• I think the bigger question is: What do we want the bigger macro-outcome to be? As a society, do we save as many lives as possible? Do we want to save individual lives? What we want as a society? Because I think the triage thing works if we want to save as many lives as possible. If we don't want to do that, and rather save first responders or whatever, then that is a different discussion. So first and foremost, what do we want the outcome to be, and that will guide us in the direction we want to go and what values we want to place on the choices we want to make and that sort of thing. (OP17) | • . . .For me, dignity and respect are the two values that come to mind. . . (OP5)<br>• Ok. I mentioned the dignity, respect. . . Those stand out in my mind. (OP21) |

did not agree because he considered that this was a political problem and not a health system problem.

> . . .I definitely agree. . .about indigenous communities. They have been mistreated for so long and still continue to be mistreated. That is one community where we should prioritize their health and well-being. . .Do I think that we can actually get to a point where we are prioritizing them? No, because of the ongoing racism that is still happening. Although I do feel that it would be ideal, it is maybe a bit unrealistic. . . (ON1)

> These are medical issues; they are not political issues. So, as much as we might want to deal with the historical wrongs that have been suffered by First Nations people in this country, that is a political issue. There are other ways to solve it. . . (ON10)

Others wondered about the best way to resolve these injustices.

> . . .The challenge is how to deal with these marginalized people now. It will take years to change the system. And we can't change the past. So, we need to be selective and say, what can we put in the protocol that will address the current situation? That is what we need to be focusing on. . . When I look at the bottom line, I ask what is the best care at the right time for these people in this sort of situation? I don't know. (ON21)

## Conditions which may favor a greater public acceptance of these protocols

**Legitimizing the triage protocol as a health strategy or policy.** According to our participants, triage protocols could gain in legitimacy if transformed into policy or pandemic emergency legislation. This was the condition that stood out the most during the deliberation. Those policies must be simple and put in place before a crisis situation to avoid chaos.

> What I think is that you should make a protocol that is fair, and get that protocol agreed with by as many people as possible to put it into law. And then you can tweak it. Right now, we've been at it for a long time and it is patchwork at best. So, if you have something that's fair, then you can take it apart or add to it or something like that. (ON19)

**Population education about triage and importance of standardization in an extreme context.** Increasing public awareness regarding triage protocols and alternative care options for those not prioritized under such protocols is the basis for a better understanding of resource allocation under extreme conditions.

*Let's say if I had a family member that was going into ICU or a spouse. There can be some compassionate grounds. Like somebody that represents the hospital would also walk through the hospital and come up to the person and say that these are the standards that we follow.* (ON12)

Participants also recommended that prioritization be standardized across the provinces, and that the criteria be respected at all levels.

*"The protocol is established, then you don't go outside the rules; if you go outside you become emotional; so we stay within the protocol we have."* (QC2)

*We should at least explain the reason and have some documentation. Maybe through the hospital website or some other formalization of it where it says that this is the standardization that we follow so that people understand that it isn't just other people attacking them.* (ON12)

**Alternative care and support for the patient's relatives or caregivers.** Participants expressed their interest in not abandoning patients who will not be prioritized under the protocol.

*If you have to choose, but there is no immediate care, no intensive care, but alternative care as you say, I think he gets the same care, but it's certain that there may not be the respirator, but there are other alternatives to be cared for, so that he's not abandoned completely, anyway, so that there is a follow-up too. . ."* (QC15)

The participants pointed out the need for psychosocial support for the patient, family members and caregivers.

*The person would need to have a level of compassion, first and foremost. If a decision has to be made, it's tough for the doctors and nurses because they're in the firestorm. But I think it takes some compassionate people to support people around it. Maybe that's separate but I think they should explain it.* (ON12)

*Most hospitals have a social worker group within it. And they could be doing a lot of assistance with patients and going over this type of criteria with them once it's set. And from there, they can be working with the patients to help them determine whether or not they're being treated. Also, at the same time, something that obviously happens is that patients come out of the ICU.* (ON26)

## Discussion

The main results of this online DD show that most participants agree with the need to develop and implement triage protocols in the extreme context of a pandemic. They support a utilitarian, benefit-maximizing approach aimed at saving as many lives as possible. However, they also stress the importance of equity, so as not to exacerbate existing inequities in our society.

Certain criteria should be considered, such as pregnant women (specific criteria absent from the Quebec and Ontario protocols), behaviors regarding health and special attention to vulnerable people in our society. For triage protocols to be acceptable, they need to be standardized, endorsed by authorities, and made public. Although triage is necessary, patients should never be abandoned. As such, support should be offered to patients and relatives of those who would be denied access to ICU.

In several countries, results from online community surveys showed support for prioritizing patients with better chances of survival [22–26]. Participants also showed strong support for considering patient survival prognosis as a highly relevant criterion for prioritization. This confirms a greater acceptability for a utilitarian perspective, which is in accordance with triage protocols and guidelines around the world [15, 35]. Although some participants remarked that we should not limit ourselves to this approach only. This would be in line with the multi-value integrated utilitarian approach considered in COVID-19 pandemic by most experts to mitigate potential discriminatory effects towards vulnerable populations [36–38].

While the majority of participants who took part in the process were in favor of a triage protocol based on best chance of survival, a few said they were neutral or didn't know. This emphasizes an important point. In extreme situations of resource scarcity, in the opinion of the experts and participants consulted, a clear, easy-to-use and transparent emergency plan is needed to avoid catastrophe and improvisation. Being indecisive or neutral could lead decision-makers to make no choice. There are many reasons for not making a choice: fear of making a mistake in the face of uncertainty and lack of evidence, fear of the political stakes or repercussions of a decision, inability to choose in the face of a plurality of equally important values, such as respect for life, the right to equal opportunities, etc. All these reasons "paralyze" decision- making. However, not making a decision is still a choice: to face the consequences of potential chaos leading to death and moral distress for all parties involved. Taking a stance on the question of triage upstream and even during a crisis enables better planning, more consultation and the avoidance of many pitfalls.

Although healthcare systems do not base healthcare delivery and distribution of resources on assessments of personal responsibility, it is interesting to note that this argument is found in our study and elsewhere. In a recent public consultation, lay people assigned lower priority to unvaccinated patients because they were considered to have "broken the social contract" with their fellow citizens, and this became a kind of retribution for their behavior [39]. Similar results were found in public consultations about the merit value, in public surveys [23–25]. These findings probably reflect the difficult social context and frustrations experienced by communities during the pandemic.

Triage protocols in Quebec and Ontario did not give priority to pregnant women, unlike other protocols contemplated in the USA (e.g., Maryland, Massachusetts, Pennsylvania, and Utah) where additional weight was assigned to them to increase their priority [3, 40]. The online DD participants pointed out the importance of giving them priority, probably assuming that pregnant women have high social, symbolic and instrumental value (role of mother) in addition to offering the potential of another life saved. For these reasons, they were considered worthy and given higher priority than others. But how can this aspect be integrated into a triage protocol? Further research into this aspect is needed.

Now that the pandemic has subsided, it is important to draw lessons and improve the protocols that have been developed by incorporating elements that are important to the public such as support not only for patients but also for healthcare providers [2, 12]. Transparent information for patients and/or their legal representatives, could facilitate decision making and avoid misunderstandings regarding the purpose of triage protocols. Misinterpretations of their goals can lead to resentment on the part of the population [41, 42]. There is also a risk of

politicizing the issues at stake, which is why transparent communication is crucial, as participants in the online DD clearly emphasized. Many authors of this study were involved in triage protocols in one way or another, as managers, designers, clinicians or patients. During the development of the protocols, we were haunted by doubt and wondered if we were doing the right thing. The DD enabled us to make explicit and analyze in greater depth what were initially strong moral intuitions. Seeing the same concerns shared by the participants reassured us and helped us realize that our moral intuitions could be debated and find valid moral justification from people outside the healthcare system.

Our study has some limitations, including the fact that the participating public did not constitute a statistically representative sample of the communities studied. It nonetheless constituted a sample with a diversity of demographic variables of the participants that favored our qualitative analysis. The main threat to DD exercise is the risk that presenters/experts, research team members present a biased view of the information (even unconsciously), leading to biased findings. To counter this risk, we ensured that the expert presentations were made independently. Plus, the experts from Quebec and Ontario had to work together to arrive at similar content, which led to double validation of content. Finally, entrusting the facilitation to the INM meant that the research team could not interfere with the discussions or the process, leaving the participants totally free to express themselves in complete safety. During the online DD, a consensus would be stimulated but not forced. Conducting an online deliberative process can also encounter technical challenges, but in sum, the participants had the necessary conditions for this deliberative exercise to be carried out in the best way. Despite these limitations, we were able to obtain valuable insights on the subject.

## Conclusion

This study provides a portrait of the Quebec and Ontario public's perspectives on prioritization protocols for access to critical care in a pandemic context. It also sets out the considerations and conditions that they believe need to be taken into account to make such protocols more acceptable. Participants supported the idea of prioritizing patients with the best prognosis of survival under extreme conditions. This was accompanied by other values such as equity, intergenerational equity, equality, solidarity, responsibility, transparency, dignity, and respect, among others. Most of the participants considered that prioritization protocols are necessary to try to reduce arbitrariness in decision making and to facilitate these difficult decisions made by health professionals. It is necessary to continue to explore the public's perspectives on these protocols to optimize them before a new pandemic emerges.

## Supporting information

**S1 Table. Intercoder reliability.**
(DOCX)

## Acknowledgments

To the INM for their guidance and collaboration in the development of the deliberations. To all the participants from the public for their valuable input to this study. We also want to thank Vanessa Finley Roy for revising the manuscript.

## Author Contributions

**Conceptualization:** Marie-Eve Bouthillier, Yanick Farmer, Claudia Calderon Ramirez.

**Data curation:** Marie-Eve Bouthillier, Claudia Calderon Ramirez.

**Formal analysis:** Marie-Eve Bouthillier, Yanick Farmer, James Downar, Andrea Frolic, Joseph Dahine, Gina Bravo, Audrey L'Espérance, Antoine Payot, Nathalie Orr Gaucher.

**Funding acquisition:** Marie-Eve Bouthillier.

**Investigation:** Marie-Eve Bouthillier, Yanick Farmer, Claudia Calderon Ramirez.

**Methodology:** Marie-Eve Bouthillier, Yanick Farmer, Gina Bravo, Audrey L'Espérance, Nathalie Orr Gaucher.

**Project administration:** Marie-Eve Bouthillier.

**Software:** Claudia Calderon Ramirez, Karell Laporte.

**Supervision:** Marie-Eve Bouthillier, Yanick Farmer.

**Validation:** James Downar, Andrea Frolic, Joseph Dahine, Lucie Opatrny, Diane Poirier, Gina Bravo, Antoine Payot, Peter Tanuseputro, Louis-Martin Rousseau, Vincent Dumez, Annie Descôteaux, Clara Dallaire, Nathalie Orr Gaucher.

**Writing – original draft:** Marie-Eve Bouthillier.

**Writing – review & editing:** Marie-Eve Bouthillier, Yanick Farmer, Claudia Calderon Ramirez.

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
