## [Decision Letter · Decision Letter 0]

18 Sep 2024

PONE-D-24-25281Public perspectives on COVID-19 triage protocols for access to critical care in extreme pandemic contextPLOS ONE

Dear Dr. Bouthillier,

Thank you for submitting your manuscript to PLOS ONE. After careful consideration, we feel that it has merit but does not fully meet PLOS ONE’s publication criteria as it currently stands. Therefore, we invite you to submit a revised version of the manuscript that addresses the points raised during the review process.

We look forward to receiving your revised manuscript.

Kind regards,

Rodrigo Toniol

Academic Editor

PLOS ONE

Journal requirements: 1. When submitting your revision, we need you to address these additional requirements. Please ensure that your manuscript meets PLOS ONE's style requirements, including those for file naming. The PLOS ONE style templates can be found at https://journals.plos.org/plosone/s/file?id=wjVg/PLOSOne_formatting_sample_main_body.pdf and https://journals.plos.org/plosone/s/file?id=ba62/PLOSOne_formatting_sample_title_authors_affiliations.pdf. 2. We note that the grant information you provided in the ‘Funding Information’ and ‘Financial Disclosure’ sections do not match.  When you resubmit, please ensure that you provide the correct grant numbers for the awards you received for your study in the ‘Funding Information’ section. 3. Thank you for stating the following financial disclosure:  [Corresponding author Marie-Eve Bouthillier received funding from a grant awarded by the Canadian Institutes of Health Research (CIHR). Funding Reference Number/ EG4-79448. ].  Please state what role the funders took in the study.  If the funders had no role, please state: ""The funders had no role in study design, data collection and analysis, decision to publish, or preparation of the manuscript."" If this statement is not correct you must amend it as needed. Please include this amended Role of Funder statement in your cover letter; we will change the online submission form on your behalf. 4. We note that you have indicated that there are restrictions to data sharing for this study. For studies involving human research participant data or other sensitive data, we encourage authors to share de-identified or anonymized data. However, when data cannot be publicly shared for ethical reasons, we allow authors to make their data sets available upon request. For information on unacceptable data access restrictions, please see http://journals.plos.org/plosone/s/data-availability#loc-unacceptable-data-access-restrictions.  Before we proceed with your manuscript, please address the following prompts: a) If there are ethical or legal restrictions on sharing a de-identified data set, please explain them in detail (e.g., data contain potentially identifying or sensitive patient information, data are owned by a third-party organization, etc.) and who has imposed them (e.g., a Research Ethics Committee or Institutional Review Board, etc.). Please also provide contact information for a data access committee, ethics committee, or other institutional body to which data requests may be sent. b) If there are no restrictions, please upload the minimal anonymized data set necessary to replicate your study findings to a stable, public repository and provide us with the relevant URLs, DOIs, or accession numbers. Please see http://www.bmj.com/content/340/bmj.c181.long for guidelines on how to de-identify and prepare clinical data for publication. For a list of recommended repositories, please see https://journals.plos.org/plosone/s/recommended-repositories. You also have the option of uploading the data as Supporting Information files, but we would recommend depositing data directly to a data repository if possible. Please update your Data Availability statement in the submission form accordingly. 5. In the online submission form, you indicated that [Data supporting the findings and materials are available upon reasonable request to the corresponding author. Some data are confindentials.]. All PLOS journals now require all data underlying the findings described in their manuscript to be freely available to other researchers, either 1. In a public repository, 2. Within the manuscript itself, or 3. Uploaded as supplementary information.This policy applies to all data except where public deposition would breach compliance with the protocol approved by your research ethics board. If your data cannot be made publicly available for ethical or legal reasons (e.g., public availability would compromise patient privacy), please explain your reasons on resubmission and your exemption request will be escalated for approval. 6. Please include captions for your Supporting Information files at the end of your manuscript, and update any in-text citations to match accordingly. Please see our Supporting Information guidelines for more information: http://journals.plos.org/plosone/s/supporting-information.

Reviewers' comments:

Reviewer's Responses to Questions

**Comments to the Author**

1. Is the manuscript technically sound, and do the data support the conclusions?

Reviewer #1: Yes

Reviewer #2: Yes

2. Has the statistical analysis been performed appropriately and rigorously? 

Reviewer #1: N/A

Reviewer #2: Yes

3. Have the authors made all data underlying the findings in their manuscript fully available?

Reviewer #1: Yes

Reviewer #2: Yes

4. Is the manuscript presented in an intelligible fashion and written in standard English?

Reviewer #1: Yes

Reviewer #2: Yes

5. Review Comments to the Author

Reviewer #1: The manuscript presents a fundamental discussion on contemporary bioethical issues and contributes to the deepening of issues related to decision-making in critical contexts in health emergencies. Considering the multiple factors and controversies surrounding the access and distribution of healthcare in a scenario of limited resources.

Below are some suggestions for improving the quality of your article, in line with the journal's criteria:

The demographic analysis of the data presented in the "Results" section could be improved. For example, I suggest that when mentioning demographic categories specific to Canada, such as "visible and non-visible minority", a footnote be inserted explaining what this means in the Canadian census (e.g. who is included in this category). Considering that this is an article for an international audience.

In additon, although table 3 mentions the gender of the participants, it is not described in the body of the text. I suggest including this and mentioning that there were no gender diverse participants.

Finally, even though the results presented are consistent and show the complexity and relevance of the subject through the survey participants' responses, the discussion section could be developed further. It would be interesting for the authors of the article to include a paragraph on the bioethical controversies involved in the application of triage protocols, expressed by the opinion of the participants categorized as "neutral or Don't know". The ethical issues seem to be associated with the variety of values and principles of equality mentioned by the DD participants.

Reviewer #2: First of all I want to thank the team for submitting the article and presenting the results in such an adequate manner. I am happy to say that the article is rich and should be published. As a researcher that has witnessed the difficulties faced by healthcare workers during the covid 19 pandemic, specially in the process of decision making and resource allocation, I’m glad to see this discussion being made outside of hospitals.

Besides the conclusions presented I think one of the great merits of this article is to have made a platform for discussing the ethical and moral quarrels of these decisions. The quotes presented show the innate struggles of reaching consensus on extreme situations like the ones experienced during the covid 19 pandemic.

At the same time I would provoke the writers to take a step further with its analysis. The discussion of the DD shows the tensions and dissents that were brought up when lay people tried to discuss - roughly put here - “the fairer approach” to the triage protocols during covid-19 pandemic. The results show that, again, putting in a rough manner, “the fairer approach” would be an “egalitarian, utilitarian, transparent and accountable approach”. At this point I would point out that some of the concepts highlighted by the DD are not necessarily self explanatory, they carry within them moral and ethical values. Let’s take the idea of transparency, for instance. Transparency as a democratic value is a rather recent idea, dating back to the 1990’s. The study made by Hetherington, “Guerrilla auditors: the politics of transparency in neoliberal Paraguay.” (2011, Duke Press) showcases how the idea of transparency was executed in Paraguay by a multiplication of papertrail that eventually would become an overbearing shadow for those that were willing to better understand the processes that should be made more transparent in the first place, and worse, this would be the only socially legitimate way to achieve clarity over the guerilla processes that Herington was investigating in the first place.

Anyway. As I said, this isn’t a necessity and the article works fine without these debates, but I think maybe the article would gain a lot by showcasing some of the nuances of these ideas and the different values that were inhabiting the notions of transparency, accountability, etc. I would suggest that these values are not inherent, but can only be built and legitimized in an effective manner through the exchange of views made possible by the DD. More than an exercise to produce data, I would argue the DD was an exercise on making the triage protocols fairer and this is a major accomplishment by your team of authors.

Finally, I would suggest adding a small point on how the authors have perceived the points made by the participants and if, as health workers themselves, they felt that the discussion was able to showcase some of their own worries and difficulties as decision makers.

6. PLOS authors have the option to publish the peer review history of their article (what does this mean?). If published, this will include your full peer review and any attached files.

Reviewer #1: No

Reviewer #2: **Yes: **João Balieiro Bardy

---

## [Author Response · Author response to Decision Letter 0]

24 Oct 2024

Point by point responses

Answer: We consulted the PLOS ONE style templates and adapted our manuscript to meet PLOS ONE's style requirements.

 [Corresponding author Marie-Eve Bouthillier received funding from a grant awarded by the Canadian Institutes of Health Research (CIHR). Funding Reference Number/ EG4-179448. ]. 

Answer: We reviewed the financial disclosure and added the role of the funders as recommended : 

“Corresponding author Marie-Eve Bouthillier received funding from a grant awarded by the Canadian Institutes of Health Research (CIHR). Funding Reference Number/ EG4-179448. The funders had no role in study design, data collection and analysis, decision to publish, or preparation of the manuscript.”

We included this information in our revised cover letter. 

5. In the online submission form, you indicated that [Data supporting the findings and materials are available upon reasonable request to the corresponding author. Some data are confidential.]. 

Answer: I contacted my research ethics committee at the Université de Montréal and they informed me that my ethical certification does not allow the sharing of data (even anonymized) on a public repository since the participants did not consent for large sharing of their data. Here's their reply: 

Hello Ms. Bouthillier,

Thank you for your e-mail. More and more journals are requiring this type of access, but it doesn't always correspond to the provisions set out in the protocols, or to the commitments made to participants. Looking at your project on Nagano, it appears that a secondary use of data has been foreseen, but within a very specific framework: “Your data could be shared with the other co-researchers and students of the research team of this project, who will also commit to keeping your data and everything discussed in the deliberation confidential”. Thus, making your project data publicly accessible could run counter to the commitments made to the participants at the time. Indeed, they did not consent to the wide distribution of their data. If you are uncomfortable with this public release, and know that it was not planned and was not consented to by the participants, it is ethically possible to refuse this request. However, if you wished to allow this distribution, a request could be submitted to the committee to do so, but this would potentially require several adjustments, or even a re-contact of the participants if you still have their contact details. 

I hope you find these clarifications useful,

If you'd like to discuss this in person via Teams, I'd be delighted to do so. Of course, I'm always available if you need me.

Have a nice weekend,

Yours sincerely

Marie-Alexia MASELLA, M.A.

Research Ethics Advisor

Office of Responsible Conduct in Research

Université de Montréal

Website: crr.umontreal.ca

Conclusion : For these reasons, we will provide access on request to certain parts of the data as long as this complies with the guidelines determined by our ethical certification. Thereofre this statement would remain valid : “Data supporting the findings and materials are available upon reasonable request to the corresponding author. Some data are confidential.”

Answer: We included captions for our Supporting Information files at the end of the manuscript and made sure that the in-text citation match accordingly.

Answer: We reviewed our reference list to ensure it is complete and correct. We did not include nor retracted any references.

5. Review Comments to the Author

Reviewer #1: The manuscript presents a fundamental discussion on contemporary bioethical issues and contributes to the deepening of issues related to decision-making in critical contexts in health emergencies. Considering the multiple factors and controversies surrounding the access and distribution of healthcare in a scenario of limited resources.

Below are some suggestions for improving the quality of your article, in line with the journal's criteria:

The demographic analysis of the data presented in the "Results" section could be improved. For example, I suggest that when mentioning demographic categories specific to Canada, such as "visible and non-visible minority", a footnote be inserted explaining what this means in the Canadian census (e.g. who is included in this category). Considering that this is an article for an international audience.

Answer : Thanks for pointing this. We added a footnote stipulating : 

“In Canada, the Employment Equity Act defines visible minorities as "persons, other than Aboriginal peoples, who are non-Caucasian in race or non-white in colour". The visible minority population consists mainly of the following groups: South Asian, Chinese, Black, Filipino, Arab, Latin American, Southeast Asian, West Asian, Korean and Japanese. The 'Multiple visible minorities' category includes persons who provided two or more groups designated as visible minorities.” 

For the sake of clarity, we also added two sentences (line 161-163) :

 “Participants were able to self-identify as belonging to any one of the visible minority groups, a category composed of multiple visible minority groups, or not belonging to a visible minority. These groups are mutually exclusive.” 

And we made changes in the Table 3, we think this presentation is clearer. Nothing substantial has been changed, just the name of one categories : we change «Ethnocultural groups» for «Self-reported visible minority groups», we think it is more precise. 

In addition, although table 3 mentions the gender of the participants, it is not described in the body of the text. I suggest including this and mentioning that there were no gender diverse participants.

Finally, even though the results presented are consistent and show the complexity and relevance of the subject through the survey participants' responses, the discussion section could be developed further. It would be interesting for the authors of the article to include a paragraph on the bioethical controversies involved in the application of triage protocols, expressed by the opinion of the participants categorized as "neutral or Don't know". The ethical issues seem to be associated with the variety of values and principles of equality mentioned by the DD participants.

Answer: Thank you for this comment. We reviewed all the transcripts in NVivo related to «neutral» or «I don’t know», and we added our interpretation of the bioethical controversies in the discussion line 376-387 :

“While the majority of participants who took part in the process were in favor of a triage protocol based on best chance of survival, a few said they were neutral or didn't know. This emphasizes an important point. In extreme situations of resource scarcity, in the opinion of the experts and participants consulted, a clear, easy-to-use and transparent emergency plan is needed to avoid catastrophe and improvisation. Being indecisive or neutral could lead decision-makers to make no choice. There are many reasons for not making a choice: fear of making a mistake in the face of uncertainty and lack of evidence, fear of the political stakes or repercussions of a decision, inability to choose in the face of a plurality of equally important values, such as respect for life, the right to equal opportunities, etc. All these reasons “paralyze” decision- making. However, not making a decision is still a choice: to face the consequences of potential chaos leading to death and moral distress for all parties involved. Taking a stance on the question of triage upstream and even during a crisis enables better planning, more consultation and the avoidance of many pitfalls.”

Reviewer #2: First of all I want to thank the team for submitting the article and presenting the results in such an adequate manner. I am happy to say that the article is rich and should be published. As a researcher that has witnessed the difficulties faced by healthcare workers during the covid 19 pandemic, specially in the process of decision making and resource allocation, I’m glad to see this discussion being made outside of hospitals.

Besides the conclusions presented I think one of the great merits of this article is to have made a platform for discussing the ethical and moral quarrels of these decisions. The quotes presented show the innate struggles of reaching consensus on extreme situations like the ones experienced during the covid 19 pandemic.

At the same time I would provoke the writers to take a step further with its analysis. The discussion of the DD shows the tensions and dissents that were brought up when lay people tried to discuss - roughly put here - “the fairer approach” to the triage protocols during covid-19 pandemic. The results show that, again, putting in a rough manner, “the fairer approach” would be an “egalitarian, utilitarian, transparent and accountable approach”. At this point I would point out that some of the concepts highlighted by the DD are not necessarily self explanatory, they carry within them moral and ethical values. Let’s take the idea of transparency, for instance. Transparency as a democratic value is a rather recent idea, dating back to the 1990’s. The study made by Hetherington, “Guerrilla auditors: the politics of transparency in neoliberal Paraguay.” (2011, Duke Press) showcases how the idea of transparency was executed in Paraguay by a multiplication of papertrail that eventually would become an overbearing shadow for those that were willing to better understand the processes that should be made more transparent in the first place, and worse, this would be the only socially legitimate way to achieve clarity over the guerilla processes that Herington was investigating in the first place.

Anyway. As I said, this isn’t a necessity and the article works fine without these debates, but I think maybe the article would gain a lot by showcasing some of the nuances of these ideas and the different values that were inhabiting the notions of transparency, accountability, etc. I would suggest that these values are not inherent, but can only be built and legitimized in an effective manner through the exchange of views made possible by the DD. More than an exercise to produce data, I would argue the DD was an exercise on making the triage protocols fairer and this is a major accomplishment by your team of authors.

Answer: We're delighted to hear your thoughts on the subject and to receive your positive feedback on our project. We believe, as you do, that the DD process helps to get the word out, thereby improving protocols and making them fairer. We agree that these values are not inherent, but can only be built and legitimized in an effective manner through the exchange of views made possible by the DD. Thank you.

Finally, I would suggest adding a small point on how the authors have perceived the points made by the participants and if, as health workers themselves, they felt that the discussion was able to showcase some of their own worries and difficulties as decision makers.

Answer : Thank you for inviting us to commit ourselves more to the study. At the end of the discussion, just before the limits of the study, we add the following few sentences, lines 415-421.: 

“Many authors of this study were involved in triage protocols in one way or another, as managers, designers, clinicians or patients. During the development of the protocols, w

---

## [Editor Report · Decision Letter 1]

12 Nov 2024

Public perspectives on COVID-19 triage protocols for access to critical care in extreme pandemic context

PONE-D-24-25281R1

Dear Dr. Bouthillier,

We’re pleased to inform you that your manuscript has been judged scientifically suitable for publication and will be formally accepted for publication once it meets all outstanding technical requirements.

Kind regards,

Rodrigo Toniol

Academic Editor

PLOS ONE

---

## [Editor Report · Acceptance letter]

19 Nov 2024

PONE-D-24-25281R1 

PLOS ONE

Dear Dr. Bouthillier, 

I'm pleased to inform you that your manuscript has been deemed suitable for publication in PLOS ONE. Congratulations! Your manuscript is now being handed over to our production team.

Kind regards, 

on behalf of

Dr. Rodrigo Toniol 

Academic Editor

PLOS ONE